# A Distance-Field-Based Pipe-Routing Method

**DOI:** 10.3390/ma15155376

**Published:** 2022-08-04

**Authors:** Shyh-Kuang Ueng, Hsuan-Kai Huang

**Affiliations:** 1Department of Computer Science and Engineering, National Taiwan Ocean University, Keelung City 202, Taiwan; 2CoreTech System Co., Ltd., Zhubei City 302, Taiwan

**Keywords:** pipe routing, distance field, shortest path, visualization in manufacturing

## Abstract

Pipes are commonly used to transport fuels, air, water, gas, hydraulic power, and other fluid-like materials in engine rooms, houses, factories, airplanes, and ships. Thus, pipe routing is essential in many industrial applications, including ship construction, machinery manufacturing, house building, laying out engine rooms, etc. To be functional, a pipe system should be economical while satisfying spatial constraints and safety regulations. Numerous routing algorithms have been published to optimize the pipe length and the number of elbows. However, relatively few methods have been designed to lay out pipes which strictly meet the spatial constraints and safety regulations. This article proposes a distance-field-based piping algorithm to remedy this problem. The proposed method converts the workspace into a 3D image and computes a distance field upon the workspace first. It then creates a feasible space out of the workspace by peeling the distance field and segmenting the 3D image. The resultant feasible space is collision-free and satisfies the spatial constraints and safety regulations. In the following step, a path-finding process, subjected to a cost function, is triggered to arrange the pipe inside the feasible space. Consequently, the cost of the pipe is optimized, and the pipe path rigidly meets the spatial constraints and safety regulations. The proposed method works effectively even if the workspace is narrow and complicated. In three experiments, the proposed method is employed to lay out pipes inside an underwater vehicle, a machinery room, and a two-story house, respectively. Not only do the resultant pipes possess minimal costs, but they also meet the spatial constraints and safety regulations, as predicted. In addition to developing the routing procedure, we also design a visualization subsystem to reveal the progression of the piping process and the variation of the workspace in the run time. Based on the displayed images, users can therefore evaluate the quality of the pipes on the fly and tune the piping parameters if necessary.

## 1. Introduction

Pipes, hoses, electric circuits, and cables are commonly used to transport fuels, water, air, hydraulic power, control signals, and electric currents in ships, factories, engine rooms, houses, etc. [1]. Pipe routing is thus a necessity for laying out factories, ship building, engine construction, house design, and other industrial applications [2,3,4,5,6]. A functional pipe system must connect the terminals, avoid obstacles, be away from hazards, and satisfy various spatial constraints and safety regulations. Furthermore, its construction, operational, and maintenance costs should be optimized to meet a financial budget [1]. Hence, designing pipe systems is difficult and becomes harder if the workspaces are narrow, complex, and filled with various equipment.

The geometric shapes, complexities, and attributes of workspaces vary on a case-by-case basis, and these factors influence pipe routing in many ways. There is rarely a generic rule for designers to follow when conceiving their pipe designs. However, the safety regulations and spatial constraints which should be met by a pipe have been well studied and published in the literature. They can be briefly summarized as follows [1,7,8]:The pipe path must be collision-free and connect the sources and destinations.The pipe length and number of elbows should be minimized. The pipe path should be straight or rectangular.The pipe path should be close to the walls, floors, ceilings, obstacles, and other pipes such that pipe racks are easier to build.The gaps between the pipe and other pipes and the space separating it from obstacles should be wide enough to allow for accessibility.The pipe must be away from hazards for safety concerns. (For example, a fuel pipe should not be close to thermal devices or electrical cables.)The pipe should not cross or occupy free spaces reserved for humans and machinery.

Conditions 1 and 2 are set for producing a functionable and low-cost pipe. These two conditions can be met if Dijkstra’s shortest path method or the A* algorithm is used to lay out the pipe [1,7,8,9]. Optimizing the total cost of a group of pipes is NP-hard and can be approximated by heuristic methods, including the particle swarm intelligence, ant colony, and genetic algorithms [3,4,10,11,12,13].

Conditions 3–6 are mutually conflicting and are hard to be explicitly satisfied by the pipe. Under condition 3, the spatial gaps between the pipe and the obstacles should be small such that creating the pipe racks is easier. However, these metrics are required to be wide in conditions 4 and 5 for the sake of accessibility and maintenance. Condition 6 prohibits the usage of a free space by the pipe, even though the free space contains no equipment. To meet conditions 3–6, many piping algorithms rely on specialized cost functions to guide the direction and position of the pipe. Under the supervision of these cost functions, the pipe will receive penalties if it is close to hazards and win advantages if it is attaching to unharmful obstacles and pipes [1,5,10,13,14,15]. Nonetheless, these cost functions are usually heuristic, flexible, and numerical but not rigid. As a result, being guided by these cost functions, the pipe may meet conditions 3–6 in some cases and violate them in other situations. Furthermore, these conditions are to be approximately satisfied by the pipe because of the intrinsic characteristics of these cost functions. The spatial and safety gaps are not deterministically bounded, and the quality of the pipe is hard to predict beforehand.

In this article, a distance-field-based method is proposed for arranging pipes in narrow and complex workspaces such that not only are the costs of the resultant pipes minimized but also the spatial constraints and safety regulations are strictly satisfied by the pipes. The key steps of the proposed method can be briefly described as follows: First, by carrying out a voxelization computation, the proposed method splits the problem domain into a 3D image of voxels. A distance field [16,17] is then calculated in the free space to record the minimum distances between the free-space voxels and the obstacles. In the following step, the proposed algorithm creates a feasible space which strictly meets conditions 3–6 by peeling consecutive layers of the distance field and segmenting the 3D image. A revised Dijkstra method is then employed to lay out the pipe inside the feasible space such that the cost of the pipe is minimized and the spatial constraints and safety regulations are rigidly satisfied by the pipe.

Recently, a researcher proposed building immersive virtual reality tools [18] and expert systems for pipe routing [3,5,12]. In these systems, real-time visualization is incorporated into pipe-routing programs to help the designers lay out pipes and evaluate the results. We follow this principle and develop a graphic module for revealing the progression of the piping process. As a result, the variation of the workspace, intermediate results, and pipes under construction are shown in real time. The designers can evaluate the quality of the pipes on the fly and modify the routing parameters if necessary.

## 2. Related Work

Piping algorithms can be categorized into two major groups, the automatic methods and the semi-automatic ones [1]. The former group includes deterministic and heuristic methods [3,4,5,6,7,8,9,10,11,19], while the latter is mainly composed of expert systems and immersive virtual reality procedures [2,18]. Deterministic piping procedures always produce the optimum solution when routing a single pipe. On the other hand, heuristic algorithms are powerful for optimizing the arrangement of multiple pipes. The semi-automatic methods follow the human-in-the-loop principle. Experts’ knowledge and experiences are encoded as rules or templates and referenced for decision-making and solution initialization. It is not unusual that interactive human–machine interfaces are incorporated into these systems to enhance their computational efficiency and productivity.

Since Lee published his piping algorithm in [20], many automatic and semi-automatic pipe-routing methods have been proposed based on his approach [1]. In Lee’s work, the problem domain is divided into cells, and the pipe path is routed upon the cells. His domain-decomposition method has had profound influences on later researches, including [2,3,4,5,7,8,12,13,14,15] and this work. In [12], Kim et al. propose to split the domain into regions first. They then insert intermediate points inside the regions according to the characteristics of the domain (a truck engine). Finally, they build a graph upon the domain and compute the pipe path by using Dijkstra’s algorithm. In another work [14], Liu revises Lee’s method and uses the underlying regular grid to construct a rectilinear visibility graph such that collisions are avoided and the computation is sped up. A similar approach to representing the problem domain by using a graph can be found in the work of [5].

In [15], Jiang and Zhang decompose the workspace by using an adaptive octree structure. In their work, they first divide the workspace into eight cells (octants). Following that, the cells, intersecting with obstacles, are recursively split into smaller octants until their sizes reach a predefined limit. Thus, the free space and obstacles can be represented by octants of varying sizes. Unlike traditional space partition methods, their approach performs sub-divisions in not only the 3D space but also the 2D and 1D spaces such that the resultant cells fit the free space better and the utilization of the free space is higher. In this research, we revise Lee’s method and decompose the problem domain into a 3D image first. As a result, the workspace can be easily regionalized and modified. Following that, using image processing techniques and distance information, we construct a graph to represent the free space reserved for routing pipes. This graph is collision-free and is away from the hazards, and the spatial constraints and safety regulations are always rigidly satisfied.

The problem domains of pipe routing vary on a case-by-case basis, and the applications of the pipes are very different [2,3,4,5,6,7,15,18,19]. Experts’ experiences and domain knowledge are valuable for routing pipes. In [2], Kang et al. present an expert system for routing pipes on ship decks [2]. In their work, the researchers integrate designers’ experiences, CAD tools, and pipe-routing algorithms into a piping program. Their system aims to improve the efficiency of pipe-design procedures for surface ships. Test results showed that their expert system is more productive than manual design processes. In [18], the authors propose to build an immersive virtual reality (VR) system to assist designers on pipe routing. The problem domain is first represented in a virtual space; and VR toolkits are supported by the system for laying out the pipes in the virtual space. The intermediate results of the piping process and the workspace are displayed via a graphical user interface (GUI) such that the designers can interactively modify their designs by using the toolkits. Experimental results prove that their VR system greatly improves the qualities and efficiencies of piping processes. In this work, we also bundle a visualization module with the pipe-routing program to reveal the progression of the piping process as well as the variation of the problem domain. Hence, the users can select routing parameters to produce better pipe paths. Our visualization module combines surface and volume rendering and can generate better portraits of the workspace and pipes.

Some piping problems are so domain-specific that experts’ knowledge has dominant influences on the topology of the underlying pipes. For example, in [19], Yue et al. propose a pipe-routing algorithm dedicated to subsea production systems. Based on their domain knowledge, they invent special domain representation methods to accommodate the facilities of underwater production systems. As a result, their pipe-routing procedure can be used to lay out multi-layer star-tree and multi-layer star subsea pipe systems under the influences of seabed terrain, existing pipes, and other obstacles. Test results reveal that the costs of pipes are reduced, especially for those pipes with the multi-layer star-tree topology.

Deterministic algorithms produce the optimum solution when routing a single pipe. However, the optimization of multiple pipe routing is NP-hard and is intractable for deterministic algorithms. Many researchers have proposed to employ heuristic methods for scheduling the orders of the pipes and branches first and then use deterministic approaches to lay out individual pipes later. Other researchers propose to employ heuristic algorithms not only to schedule the piping process but also to route individual pipes. In [4], Jian et al. present an improved ant-colony-optimization (ACO) algorithm for routing pipes in surface ships. In their method, they first design an ACO procedure to lay out simple pipes. They then employ multiple groups of ants and perform permutation upon the pipes to optimize the routing of multiple pipes with multiple terminals. In [10], another ACO method is utilized to arrange pipes for ships. The approach is composed of two key stages. In the first stage, a heuristic method is employed to generate the initial paths of the pipes. In the second stage, an ACO method is applied to improve the pipe paths. In [19], a min-max ant colony method is employed to optimize pipe paths even though the underlying topology of the pipe system is domain-specific and has been decided earlier.

In addition to ACO methods, genetic algorithms (GA) are commonly used for laying out multiple pipes too. In [12], Kim et al. represent pipe routes by using chromosomes and design a fit function to optimize the pipe length and the number of elbows. To increase the efficiency of their GA program, they revise the crossover and mutation operators. They also employ the expertise of experienced designers and incorporate their knowledge into their GA to ensure that their piping program is capable of routing pipes for truck engines. In [3], Asmara and Nienhuis combine deterministic and non-deterministic algorithms to form a pipe-routing system. In their system, a deterministic Dijkstra’s algorithm is employed to lay out optimal paths for individual pipes, while particle swarm optimization methods are used to schedule branches and pipes. Our research focuses on developing data structures and algorithms for pipe routing such that spatial constraints and safety regulations are strictly followed by the resultant pipes. Thus, scheduling pipes and branches is out of the scope of this article. However, we do follow the suggestions given by published papers and experts for ordering pipes and branches [3,13]. These ideas will be presented in the following section.

Using fit functions, objective functions, and cost functions to guide pipe paths is very common in pipe-routing algorithms. In [13], Dong and Bian design an objective function to guide the direction of their A* searching process. Their objective function aims to reduce the pipe length and the number of bends and to attach the pipe to walls, equipment surfaces, and other pipes. In [4], the authors propose a fit function to guide the search directions of their ant colony. The goals of their fit function are similar to those of [13]. Fan et al. include penalties in their objective function [10]. Their objective function regards pipe length, elbows, and positions in forbidden regions as negative factors, and their ACO process optimizes the objective function to meet spatial constraints and to avoid collisions. In [8], Ando and Kimura invent a method, based on specialized cost function, to optimize pipe paths and elbows of varying sizes. In this work, we also rely on a cost function to reduce the pipe length and number of bends. Compared with other piping algorithms, our cost function is simple and easy to implement. We extract a feasible space from the workspace before routing a pipe, and therefore the spatial constraints and safety regulations are automatically guaranteed to be satisfied. Our cost function does not need to include penalties or advantageous terms to guide the direction and position of the pipe.

The rest of this paper is organized as follows. In Section 3, the proposed piping algorithm is described in detail, including the creation of the feasible space, pipe-routing computation, and visualization method. Test results are given in Section 4. Three experiments are conducted to verify the efficacy of the proposed algorithm by routing pipes in three different workspaces of high geometrical complexities. These cases are presented and discussed separately in Section 4. The characteristics of the proposed algorithm and comparisons with the related work are presented in Section 5. The pros and cons of the proposed algorithm are analyzed in Section 5 too. This article is concluded in Section 6.

## 3. Materials and Methods

In this section, the proposed piping method will be presented, including its flowchart and key computational steps. Each computational step is described in detail separately. The pseudo-codes of the algorithms dedicated to routing the pipes and computing the distance field are contained in Appendix A and Appendix B. In addition, two illustrations are used to demonstrate the progression of the distance field computation and pipe-path routing. These figures are also contained and annotated in this section.

### 3.1. Overview of the Proposed Method

The flowchart of the proposed piping method is shown in Figure 1. First, the pipe-routing program retrieves the geometric information of the problem domain from disk files. The problem domain consists of obstacles and free space, represented in a polygon-based format. In the following step, the problem domain is split into voxels and becomes a 3D image. A distance field is then computed in the free space. This distance field records the distances between the voxels of the free space and the obstacles. A pipe is then chosen from the pipe pool and serves as the target of the routing process. In the fourth step, a feasible space is extracted from the free space. This feasible space strictly satisfies the spatial constraints. The free space is then further revised such that the safety regulations are also met. In the subsequent step, the target pipe is constructed inside the feasible space by using a revised Dijkstra’s shortest-path algorithm. Once the routing computation of the target pipe is completed, the target pipe is regarded as a new obstacle, and both the free space and the distance field are updated. Following that, another round of the piping process is conducted to arrange the next pipe. When all the pipes have been constructed, the geometric representations of the problem domain and the pipes are output to disk files. For the sake of clarity, the pseudo-codes of the proposed piping algorithms are presented in Appendix A.

The key steps of the proposed piping method will be presented in the rest of this section, including the voxelization process, distance field computation, feasible space extraction, and pipe-path searching. How these computations influence the efficiency and quality of the pipe-routing process are also discussed. Conventional piping methods lack the capability to reveal the progression of the piping process. In the proposed piping procedure, a graphical module is developed for displaying the obstacles, distance field, feasible space, and pipe paths on the fly. Hence, the users can evaluate the quality of the pipe-routing process and modify the controlling parameters if necessary. This graphic module is described in this section too.

### 3.2. Voxelization and Distance Field Computation

As shown in Figure 2a, a problem domain may contain machinery, walls, floors, ceilings, etc. These elements are obstacles, and no pipe can penetrate them. The remaining space forms the free space and is available for routing pipes. In the initialization stage, the proposed piping algorithm converts the problem domain into a 3D image of voxels. Hence, the free space can be quickly formed inside the problem domain and easily modified in the later computations.

In the voxelization task, we enclose the problem domain by using a bounding box first. The bounding box is then split into voxels by using a regular grid [21]. The voxel width is smaller than the radius of the thinnest pipe, and hence the cross-section of any pipe is wider than the voxel width. Each voxel is given a specific intensity to reveal its original owner. For example, the intensity of a free-space voxel is set to 999, while the intensity of an equipment’s voxel is set to the equipment’s ID. Thus, the problem domain becomes a 3D image, and we can divide the workspace into subregions based on voxel intensities and manipulate the free space by using image processing operators.

Following the voxelization process, we invoke a numerical procedure to compute the shortest distance from each free-space voxel to the obstacles. This distance function is governed by the following Eikonal equation [16,17]:(1)$$\begin{array}{l} {\left(\frac{\partial u}{\partial x}\right)}^{2}+{\left(\frac{\partial u}{\partial y}\right)}^{2}+{\left(\frac{\partial u}{\partial z}\right)}^{2}=1,\\ u(x,y,z)=0\;in\;S. \end{array}$$
where *u* is the distance function to be solved, and *S* represents the surfaces of the obstacles, for which the distance function is set to zero.

In this research, we use the revised fast marching method (RFMM) proposed in [17] to compute *u* for Equation (1). The procedure of the RFMM can be described as follows: First, we find the voxels in the interiors of the obstacles and give them a negative distance to assert that these voxels are excluded from the pipe-routing computations. In the next step, the voxels of the free space are grouped into a set, which is called FAR. Initially, their distances are set to infinity. Following that, the voxels on the obstacles’ surfaces are searched and kept in a set called DONE. These voxels are labeled as visited, and their distances are set to zero, as shown in Figure 2b. In the following step, the voxels which are in FAR and are adjacent to DONE are removed from FAR and kept in another set, called CLOSE. Their distances are computed by solving Equation (1), as shown in Figure 2c. In this image, the interior voxels of the obstacles are rendered in gray color, while the voxels in DONE, CLOSE, and FAR are shaded in blue, green, and white colors, respectively.

After creating these three sets, the voxel that is in CLOSE and possesses the shortest distance is removed from CLOSE and inserted into DONE. Hence, DONE is enlarged. After that, those voxels which are in FAR and are adjacent to this newly visited voxel are removed from FAR and inserted into CLOSE, and their distances are updated by solving Equation (1) numerically. Thus, the scope of FAR is shrunk. These set operations and distance calculations are repeated until all voxels are moved from FAR to CLOSE and then from CLOSE to DONE. The example in Figure 2d displays the configuration of the problem domain after two layers of the distance field have been computed.

#### Distance Computation Method

To compute the distance of a voxel in CLOSE, the RFMM converts the non-linear differential equation of Equation (1) into a quadratic polynomial by first using the distances of its adjacent neighbors in DONE. The polynomial can be formulated as follows:(2)$$\begin{array}{l} au^{2}+bu+c=0,\\ u=\frac{-b\pm \sqrt{b^{2}-4ac}}{2a}. \end{array}$$
where variable *u* stands for a possible distance of the current voxel, and *a*, *b*, and *c* are the coefficients of the polynomial. These coefficients result from approximating the partial derivatives of Equation (1) by using forward and backward difference methods. The methods for determining *a*, *b*, and *c* and solving for *u* are given in Appendix B.

The two roots of this polynomial are then computed by using the quadratic formula of Equation (2). Once the roots have been calculated, the voxel’s original distance is compared with the maximum root. If the maximum root is smaller, the voxel’s distance is set to the maximum root. Otherwise, the voxel keeps its original distance. The detailed steps, pseudo-code, and formal proof of the RFMM, including the manipulation of the sets DONE, CLOSE, and FAR and the calculations of the coefficients *a*, *b*, and *c*, can be found in [17]. We omit these materials in this article and focus on the methodologies employed for meeting spatial constraints and safety regulations.

The distance field forms a level set. We can divide it into a series of layers as if we were peeling an onion. The voxelization process converts the problem domain into a 3D image, and thus we can manipulate it by using erosion, dilation, and breadth-first search. Based on these features, we extract the feasible space from the problem domain and confine the pipe routing there to ensure that the spatial constraints and safety regulations are automatically satisfied.

### 3.3. Feasible Space Creation

When building a pipe, we must first ensure that the pipe connects the terminals and is collision-free from the obstacles. We should then minimize its length as well as the number of bends. In addition, the pipe has to meet the spatial constraints and safety regulations such that its construction, usability, maintenance, and operational safety can be guaranteed. Taking so many factors into consideration simultaneously while arranging the pipe is challenging. In this research, we propose to generate a feasible space in the free space which contains no obstacles and intrinsically satisfies the spatial constraints and safety regulations first. The pipe path is then routed inside this space only, and thus detecting collisions and checking for violations of spatial constraints and safety regulations becomes unnecessary. Therefore, the complexity of the piping process is greatly simplified.

#### 3.3.1. The Primitive Feasible Space

There are two major purposes of the spatial constraints. The first one is to force the pipe to be close to the obstacles (including walls, ceilings, floors, and existing pipes) such that the construction of the pipe racks is easier and less free space is consumed. The second purpose is to preserve reasonable gaps between the pipe and the obstacles so that the installation, operation, and maintenance of this pipe will be convenient. Obviously, these two purposes conflict and are hard to achieve at the same time.

To resolve this conflict, we extract a feasible space from the free space by using the distance information kept in the free-space voxels. First, starting from the surfaces of the obstacles, we perform a breath-first search to collect those free-space voxels whose distances *u(x,y,z)* satisfy the following constraints:(3)$$d_{\min}\leq u(x,y,z)\leq d_{\max}$$

The variables *d_min_* and *d_max_* are the lower and upper bounds of the distance values of the feasible space, respectively. They are determined as follows.
(4)$$\begin{array}{l} d_{\min}=r+\varepsilon_{1},\\ d_{\max}=r+\varepsilon_{2},\\ 0\leq \varepsilon_{1}<\varepsilon_{2}. \end{array}$$
where *r* is the radius of the pipe, and *ε_1_* and *ε_2_* are the minimum and maximum allowable gaps between the pipe and the obstacles, respectively. These two parameters are adjustable, based on the attributes of the pipe. By tuning these two parameters, the designers can ensure that the feasible space is wide enough for arranging the pipe and that the pipe is neither too close to the obstacles nor too far from the obstacles.

An example is displayed in Parts *a* and *b* of Figure 3. The distance field is composed of five level sets with distances equal to 0, 1, 2, 3, and 4, as shown in Figure 3a. Assuming that the radius of the pipe is 0.5, and *ε_1_* and *ε_2_* are set to 0.5 and 1.5, the bounds *d_min_* and *d_max_* are set to 1 and 2 based on Equation (4). Thus, those voxels with distances within this range are included in the feasible space (the green and yellow voxels), as shown in Figure 3b.

#### 3.3.2. The Revised Feasible Space

By following the rules of Equation (3), the resultant feasible space meets the spatial constraints. However, it may fail to satisfy the safety regulations and may exclude some of the pipe terminals. For example, as shown in Figure 3b, the source (red star) and the destination (blue star) are disconnected from the feasible space. In addition, the feasible space is touching the hazards (purple voxels). Thus, we must revise the feasible space to solve these two problems. First, we dilate the feasible space to include all the terminals in the feasible space. Following that, we erode the feasible space such that it is separated from the hazards by a predefined distance.

The dilation starts at all the dangling terminals, which are not connected to the feasible space. We search those free-space voxels adjacent to the dangling terminals and label them as potential feasible voxels. These potential feasible voxels form small regions surrounding the dangling terminals. In the following steps, we gradually expand these regions by using breadth-first search until they touch the feasible space. Finally, we relabel all the potential voxels as feasible-space voxels and stop the dilation process.

To keep the feasible space away from the hazards, we erode those voxels which are too close to the hazards from the feasible space. First, we search the voxels next to the hazards. If they are feasible-space voxels, they are removed from the feasible space such that they will not be used in the following pipe-routing task. The erosion then continues in a breadth-first-search manner until the distance (number of layers) of the breath-first search exceeds a predefined limit. As the erosion ends, the feasible space is guaranteed to be at a safe distance from the hazards.

In Figure 3b, the source and destination of the pipe (the blue star and the red star) are not in the feasible space, and some portions of the feasible space are adjacent to hazards (purple-colored). After the dilations and erosions, the source and destination are connected to the feasible space, and the hazards are away from the feasible space. The minimum distance between the pipe and the hazards is 2 units in this example. The revised feasible space is presented in Figure 3c, where the eroded voxels are shaded with dark gray color.

### 3.4. Pipe Path Routing

In the following step, we convert the feasible space into a graph *G* = {*V*, *E*} such that we can employ Dijkstra’s method to compute the pipe path. The conversion procedure is as follows: The voxels in the feasible space form the vertex set *V*, and the adjacency relations between these voxels constitute the edge set *E*. Because the feasible space is a subset of a 3D regular grid, each vertex has at most six neighbors, residing in the ±x, ±y, and ±z directions. Thus, the pipe path will consist of only straight lines and rectangular bends.

After creating *G* and before routing the pipe, we calculate the Manhattan distances between all the pairs of terminals and find the pair of terminals that possesses the longest Manhattan distance. One of these two vertices is designated as the source, while the other one serves as the destination. In the subsequent step, the shortest path connecting these two vertices is computed by using Dijkstra’s algorithm.

Once the path-finding task has been completed, the vertices residing in the computed path are regarded as destinations, and one of the remaining terminals is treated as the new source. The path-finding procedure is then carried out to build a branch connecting this terminal to the existing path. The above computations are repeated until all the terminals have been connected to the pipe path. The pseudo-code of the pipe-routing procedure can be found in Appendix A.

#### The Cost Function

Because *G* is derived from a 3D regular grid, the resultant pipe path would contain many elbows if we merely employed Dijkstra’s algorithm to find the shortest pipe path. To reduce the bends, the following cost function is used to measure the distance between the source and the vertex which is to be visited in the next step:(5)$$\begin{array}{l} C(v_{source})=0\\ C(v_{i+1})=C(v_{i})+p(v_{i},v_{i+1}),\\ p(v_{i},v_{i+1})=\{\begin{array}{c} 1,\;if\{v_{i-1},v_{i},v_{i+1}\}\;are\;collinear,\\ 1+\omega ,\;otherwise. \end{array} \end{array}$$
where *C(v)* is the accumulated distance from the source to vertex *v*, and *p(v_i_, v_i+_*_1_*)* represents the cost of the edge connecting vertices *v_i_* and *v_i+_*_1_. Vertices *v_i−_*_1_, *v_i_*, and *v_i+_*_1_ stand for the previous, current, and next visited vertices in the path-finding calculation, respectively, and *ω* is a penalty selected by the users to punish the pipe if it makes a 90-degree turn. This cost function increases the path length by 1 + ω, instead of 1, if the path makes a bend. Hence, straight paths are preferred over elbows.

Compared with conventional piping algorithms, the proposed method employs a very simple cost function. This is because the feasible space created in the previous stage already meets the spatial constraints and safety regulations. There is no need to include any penalty or benefit terms in the cost function to repel the pipe path from the hazards or to attract the pipe path to the obstacles. The image shown in Figure 3d contains the pipe path of the 2D example. The pipe path is composed of cells of the feasible space, colored in green and yellow. The pipe path is close to the walls but is always away from the hazards. It meets the spatial constraints and the safety regulations.

### 3.5. Pipe Geometry Generation and Piping Process Visualization

The pipe path constructed by the above algorithm forms only the centerline of the pipe. To complete the work, we search for the voxels which are within *r* units of the centerline (*r* is the radius of the pipe) and label them by using the pipe’s ID. Therefore, these voxels will be owned by the pipe and will be regarded as obstacle voxels in future computations. Following that, cylinders with radii equal to *r* are generated along the centerline to produce the surface of the pipe. Finally, the pipe surface is decomposed into triangular patches and kept in an STL (STereoLithography) file.

To reveal the progression of the piping process, the proposed pipe system contains a graphical module to display the problem domain, distance fields, and pipes. The graphical module depicts pipes by using surface rendering first. It then displays the obstacles, free space, and feasible space by using volume rendering such that the internal structures of the problem domain as well as the related positions of the pipes inside the workspace can be comprehended. By using this strategy, both the depths of the pipes and their adaptation to the obstacle surfaces are illustrated.

In the volume-rendering computations, our graphical module assigns high opacities to the voxels belonging to the boundaries of the free space, feasible space, and obstacles and gives low opacities to the voxels inside these regions. Therefore, these regions are revealed, and the pipes are not occluded or blurred in the resultant images. To achieve real-time visualization, the rendering process is accelerated by using the graphics processing unit (GPU). Detailed steps of the real-time volume rendering can be found in [22].

## 4. Results

Based on the proposed pipe-routing method, we implemented a software system dedicated to routing pipes. Three experimental workspaces had been created to test the capabilities of the piping program. These workspaces are the chamber of an underwater vehicle, a machinery room, and a house. Each workspace is divided into layers and rooms by using floors, ceilings, and walls. To connect these rooms, there are holes in the walls, floors, and ceilings. Thus, pipes must go through these holes to connect their terminals located in different rooms. Multiple pipes are arranged in each workspace, and each individual pipe is composed of several terminals residing in different rooms. Thus, the pipes not only have to penetrate the holes but also have to grow branches to connect their terminals. Furthermore, the pipe diameters are varied. This makes the pipe-routing process more challenging.

### 4.1. Pipe Routing inside an Underwater Vehicle

Pipe routing is difficult in surface ships [2,3,4,5,8,10]. This task becomes even harder if the workspace is an underwater vehicle. The reasons can be explained as follows: The weight and volume of an underwater vehicle are in delicate balance [23]. The internal volume of the pressure hull is fixed after the concept design stage [23], and most of the space inside the pressure hull must be reserved to accommodate the crew and to install the facilities. The remaining space available for arranging pipes is small, and thus the pipes should be packed together and attached to walls or ceilings to save space. For the sake of survivability, the pressure hull of an underwater vehicle is usually divided into compartments by using bulkheads which can withstand high pressures and fires. To increase the utilization of space, each compartment is split into rooms by using floors and walls. As a result, the terminals of a pipe may reside in different rooms, and the pipe path must go through the holes in the bulkheads, walls, and floors to connect the terminals [23]. Thus, the complexity of pipe routing is dramatically increased in underwater vehicles.

To test the capability of the proposed piping algorithm, we purposely create a domain mimicking the internal space of the pressure hull of an underwater vehicle. The layout of the domain is shown in Figure 4a. It consists of four compartments separated by three bulkheads. Each compartment is divided into two or three decks by floors. There are holes in the bulkheads and the floors such that the spaces of all the rooms are connected. The length, width, and height of the workspace are 8.02, 2.00, and 2.00 m, respectively. This workspace is split into voxels by using a regular grid. The grid resolutions in the X-, Y-, and Z-directions are 802, 200, and 200, respectively. Thus, the shape of each voxel is 1 cm × 1 cm × 1cm, as depicted in Table 1.

#### 4.1.1. Routing the First Pipe

Two pipes are to be routed through this workspace. The positions of the terminals are specified by using the indices of the voxels. The first pipe consists of seven terminals, and its radius is 4 cm. The coordinates of the terminals are listed in Table 2. Before routing this pipe, we compute the distance field of this domain. Two levels of the distance field are displayed in Part b of Figure 4. The obstacles are rendered in gray color, while the levels of the distance field are shaded in red and green colors. The red-colored level is at a distance of 10 cm to the obstacles, while the green-colored layer is 20 cm away from the obstacles. A feasible space is then constructed inside the workspace and shown in Part c of Figure 4. The feasible space is shaded in green color in this image. The minimum and maximum distances of the feasible space are 5 and 9 cm, respectively. As the image shows, it tightly attaches to and surrounds the obstacles.

Following that, the Manhattan distances between all the terminals are computed. The Manhattan distance between terminals 1 and 2 is the longest, and thus the first branch of the pipe is aiming to connect terminals 1 and 2. The branches connecting terminals 3, 4, 5, 6, and 7 to the main pipe path are then computed. The pipe path is constructed by using Dijkstra’s algorithm and employing Equation (4) to calculate the cost of the pipe path. The penalty for generating a bend in a path is 0.5 voxel lengths.

The results are depicted in Part d of Figure 4. The walls, bulkheads, and floors are shaded in gray color by using volume rendering, while the pipe is rendered in red color by using surface rendering. Though the geometrical complexity of the workspace is high, the pipe travels the domain via the holes and successfully connects the terminals located in different rooms. Because the feasible space is within 9 cm of the obstacles, the pipe path tightly attaches to the walls, ceilings, and floors. There is plenty of space in the pressure hull for accommodating the crew, installing machinery, and constructing pathways. Thus, the resultant pipe satisfies all the spatial constraints.

#### 4.1.2. Routing the Second Pipe

After routing the first pipe, we arrange the second one. The locations of its terminals are listed in Table 3. Though the second pipe possesses fewer terminals, its terminals are farther apart and located in the front, middle, and rear compartments. We predict that the pipe path will be long and curvy.

The first pipe occupies some portions of the original free space. The revised workspace is shown in Part *a* of Figure 5. The obstacles are shaded in gray color. The distance field is then re-computed to reflect the variation of the workspace. The resultant distance field is displayed in Part *b* of Figure 5. The red-colored layer is closer to the obstacles (at a distance of 10 cm), while the green-colored level is farther from the obstacles (at a distance of 20 cm). A feasible space is then created, as shown in Part *c* of Figure 5. The minimum and maximum distances of the feasible space are 3 and 7 cm, respectively. Our system creates the longest branch first, which connects terminals 1 and 2. Following that, the branches connecting terminals 3 and 4 to the main path are constructed consecutively. The first and second pipes are shown in Part *d* of Figure 5. The pipes are rendered in red and green colors.

As the results show, the second pipe penetrates many bulkheads, walls, and floors, as its terminals are far apart from each other and located in different rooms. Secondly, the second pipe does not cross the first pipe, even though the free space is reduced. This is because the first pipe is regarded as an obstacle and excluded from the feasible space once it has been constructed. Thirdly, the second pipe attaches to the walls, ceilings, and floors, as the first pipe does. This phenomenon is caused by the scope of the feasible space, which is at most 7 cm from the obstacles. The paths of these two pipes are collision-free and meet the predefined regulations and constraints. Thus, this example verifies the effectiveness of the proposed method for creating the feasible space.

### 4.2. Routing Pipes in an Engine Room and a Two-Story House

Pipes are commonly used in machinery rooms to transport air, cooling water, and fuels. They are the veins and arteries of facilities. Hence, pipe routing is important for the layout of machinery rooms. In another test, we artificially built a machinery room as shown in Part *a* of Figure 6. This room is split into three regions by two walls (light-blue-colored). Each region contains a piece of equipment, colored in light yellow. There are holes in the walls to connect these regions. To increase the geometric complexity of the workspace, we create only one big hole in one corner of the first wall but create three layers of holes in the second wall. The holes in the top layer are large enough for any pipes to go through. The holes in the middle level allow medium- and small-sized pipes to pass. The holes in the low level are so narrow that only the thinnest pipes can penetrate them.

Three pipes are to be constructed. Each pipe has multiple terminals, located in different regions. The radii of the pipes are 5, 4, and 2 cm, respectively. The pipes must connect their terminals and attach to the floors, the walls, and the surfaces of the machines to save space. The three pipes are shaded in red, green, and blue colors. The red pipe is the thickest, while the blue pipe is the thinnest. We route the red pipe first because it is the thickest pipe and needs more space. The green pipe is then routed, followed by the blue pipe. The routing results are shown in Part *a* of Figure 6.

As the image shows, all the pipes successfully connect their terminals even though the geometrical complexity of the workspace is high, and the terminals are separated by the walls. Their paths are collision-free and are attached to the walls, floors, and facility surfaces. Only a small portion of the free space is occupied by the pipes. Thus, they strictly meet the spatial constraints.

These pipes share the big hole in the first wall, as there is no alternative gateway. However, they use different holes in the second wall because of their sizes and orders. The red pipe is the first pipe to be constructed. It occupies the best positions. Its path contains a few bends, but these bends are caused by the shapes of the facilities and the geometry of the workspace. On the other hand, the green pipe contains many bends. This is because that it can only utilize the space and holes left by the red one, and it must avoid colliding with the red pipe. Therefore, it makes some turns. Though the blue pipe is the last pipe to be built, it contains fewer bends than the green pipe. This can be explained as follows: The blue pipe is a thin pipe and can go through the small holes and narrow gaps left by the previous two pipes, and hence it has fewer bends. Its bends are mostly caused by the shapes of the facilities and the geometry of the room.

Pipes are commonly utilized in houses to transport tap water, gas, sewage, and air to maintain good living conditions. In addition to spatial constraints and safety regulations, the pipes in a house must satisfy aesthetic conditions too. Otherwise, they will degrade the values of houses. In the last test, we create a two-story house and use it as the workspace for routing pipes. Two pipes are routed to connect terminals located in all the rooms of the house to serve as water and ventilation pipes. We create holes in the walls such that the pipes can travel out of the house and then re-enter the house if necessary. However, to make the routing harder, each hole allows only one pipe to pass through.

The results are shown in Part *b* of Figure 6. The red pipe is thicker and is the first one to be constructed. As the image displays, these pipes link terminals in all the rooms and accomplish their functionalities. Both pipes attach to the ceilings, floors, and walls and are collision-free. They occupy limited space and leave plenty of spatial resources for the designers to arrange furniture and facilities. They strictly meet the spatial constraints. Furthermore, their paths travel out of the house and march along the outer surfaces of the walls and then enter the house again through the holes to connect the terminals. They do not travel in the open space and do not produce offensive visual effects on the exterior surfaces of the house. In conclusion, this experiment reveals that the proposed piping method can create pipes for buildings too and highlights a potential usage of the proposed piping method in architecture design.

## 5. Discussion

In the proposed piping method, we discretize the problem domain into a 3D image and then compute a distance field in the free space. In the following steps, we peel multiple levels of the distance field to create a feasible space and use dilation and erosion to connect dangling terminals with the feasible space and to keep the feasible space at a predefined distance from the hazards. By using these strategies, our method possesses the following advantages over conventional piping methods:The spatial gaps between the pipe and the obstacles, hazards, and other pipes are deterministically decided. Conventional piping algorithms use fit functions to direct the pipe path and hope that the spatial constraints and safety regulations are met by the pipe. As mentioned before, their fit functions may not produce the desired results because they are heuristic. Even if the pipe satisfies the spatial constraints and safety regulations, the resultant spatial gaps are not bounded or predictable.The feasible space is collision-free and satisfies the spatial constraints and safety regulations. Thus, it is unnecessary to detect collisions or to calculate the distances between the pipe and the obstacles, hazards, and other pipes at the run time.Our problem domain is composed of voxels. Modification of the workspace and creation of the feasible space are carried out in the 3D image space. They utilize mature image-processing techniques. Assuming that the image space is composed of O(n^3^) voxels, the time complexity of creating the feasible space is bounded by O(*kn^2^*), where *k* is the number of distance levels in the feasible space, and *n^2^* stands for the number of voxels in a level of the distance field. Conventional methods use polygonal facets to represent the problem domain, making extracting a region from the workspace and modifying the workspace very expensive.We assign labels to the voxels to reflect their usages and thus need not modify the geometry of the problem domain when a pipe has been built. In conventional piping methods, the geometric definition of the newly constructed pipe must be added into the obstacle set because the space occupied by the pipe can no longer be used in future computations. Subsequently, the geometrical definition of the workspace has to be overhauled, and the computational cost is increased.The workspace is composed of O(n^3^) voxels. By using A* or Dijkstra’s method, the cost of routing a pipe inside the workspace is bounded by O(n^3^). However, our piping procedure confines the pipe routing inside the feasible space which contains only O(*kn^2^*) voxels. Thus, the time complexity of the pipe routing is bounded by O(*kn^2^*). The time complexity is reduced by an order.

It is well known that Dijkstra’s algorithm is slower than the A* method at finding shortest paths in general graphs. Nonetheless, our problem domains may contain several rooms separated by walls, ceilings, and floors. These rooms are merely connected via the holes in the walls and floors. The heuristic estimate of the remaining distance of the pipe in the A* algorithm is not accurate, and it results in exhaustive search, as Dijkstra’ method does. Thus, no significant speed-up can be achieved by using the A* algorithm.

Other researchers have proposed to use iterative deepening A* (IDA*) algorithms [24,25] to search for shortest paths in graph structures. Compared with the A* algorithm, IDA* methods require less memory space, as they are conducted in a depth-first-search manner. In the aspect of computational speed, the A* and IDA* procedures possess similar worst-case time complexities. Thus, IDA* methods could not improve the time complexity of our routing process either.

Though our method is a deterministic one, the designers can adjust the feasible space by tuning the parameters of Equation (4), and hence the spatial gaps can be modified according to the intrinsic properties of the pipes. Based on a similar approach, we can adjust the distances between pipes and hazards by controlling the distance of the erosion to assure the safety of pipes. The ordering of the pipes has a great influence on the resultant pipe paths, as shown in the example of Figure 6a. Because this work focuses on methods for meeting spatial constraints and safety regulations, pipe scheduling is not concerned. We propose to include experts’ knowledge in the routing computation such that the number of elbows can be reduced. The other method is to employ AI methods [2,3,4,5,6,7,10,11,12,13] to order the pipes and use the proposed algorithm to compute the pipe paths.

The proposed piping method divides the workspace into voxels. The accuracy of the pipe routing is greatly influenced by the voxel size, including pipe lengths, spatial gaps, and distances to hazards. The accuracy is linearly proportional to the resolution of the underlying grid. Increasing the grid resolution improves the precision. However, the memory usage is enlarged too. If the main memory space is limited, we proposed to adopt an out-of-core strategy to solve the problem. First, we represent each room as an independent domain and set intermediate points in the holes. Each room is then kept in a disk file. The pipe routing is performed independently in each room. Therefore, we have to fetch the digital representation (a 3D image) of only one room into the main memory at the run time. We then create the feasible space for the current room and compute the all-pair shortest paths between the intermediate points and the terminals. Following that, the pipe paths of the current room are exported to a disk file. After completing the routing tasks for all the rooms, we fetch the pipe paths of all the rooms into the main memory and merge them according to the intermediate points to complete the pipe-routing process.

Recently, some researchers published state-searching algorithms based on the IDA* method [24,25]. These algorithms have been successfully applied in gaming applications and path-routing problems for large graphs. These IDA* methods possess the same worst-case time complexity as the traditional A* and Dijkstra’s algorithms. However, their spatial complexities are smaller by an order. In our future work, we propose to use IDA* algorithms for routing pipes in large workspaces to save memory usage.

## 6. Conclusions and Future Work

In this article, we present a pipe-routing method. The major capabilities of the proposed method can be summarized as follows: First, the costs of the resultant pipes are optimized. Second, spatial constraints and safety regulations are strictly obeyed by the pipes. Third, the progression of the piping process is revealed in real time. To accomplish these results, the proposed method decomposes the problem domain in a 3D image and computes a distance field upon the free space of the problem domain. A free space is then extracted from the distance field such that the spatial constraints and safety regulations are automatically satisfied. In the following step, a revised Dijkstra’s method is employed to compute the pipe path such that the pipe’s cost is reduced. In addition, the piping procedure contains a graphical module which is capable of displaying the workspace, distance field, and pipes in real time by using surface and volume rendering. Three experiments were conducted to test the proposed piping method. In the experiments, the proposed piping procedure produced decent results in different workspace, and all the goals of piping were met. Its efficacy has been verified by these experiments.

The scheduling of the pipes plays an important role in the quality of the pipe system. We propose to utilize experts’ experience and AI methods for scheduling pipe routing while employing our piping algorithm to route individual pipes. Memory usage is another concern for the proposed piping method. If the main memory space is not sufficient to carry out our high-precision pipe-routing process, we suggest that the piping process should be carried out in an out-of-core manner. In the out-of-core piping process, we divide the problem domain into rooms and lay out pipes in each room first. The pipes are then merged to form the final pipe system. IDA* algorithms require less memory than A* and Dijkstra’s methods in path finding. In future work, we intend to employ an IDA* searching algorithm for calculating pipe paths to further reduce memory usage in pipe routing.

## Figures and Tables

**Figure 1 materials-15-05376-f001:**
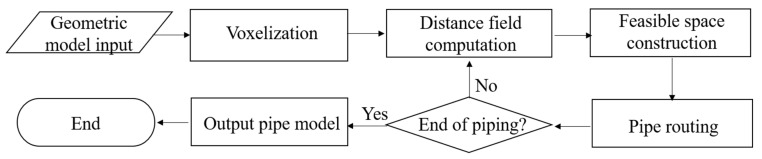
Flowchart of the proposed piping method.

**Figure 2 materials-15-05376-f002:**
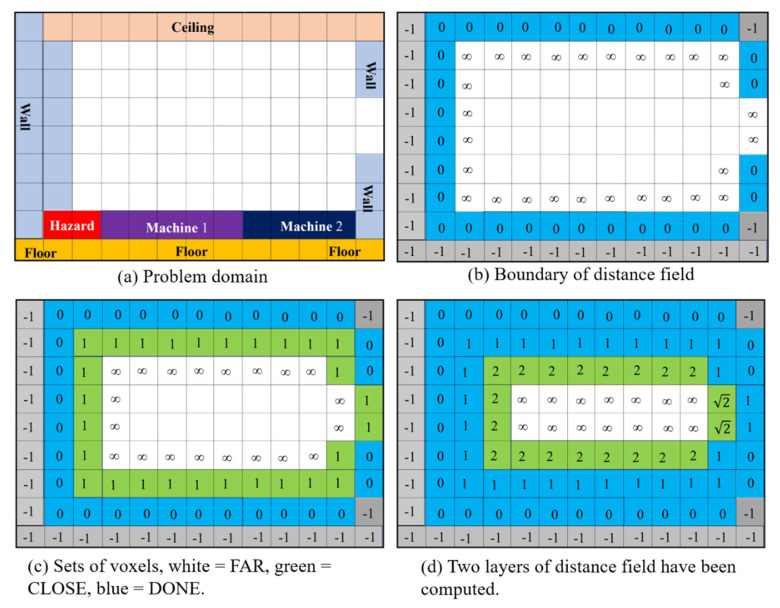
Distance field computation in a 2D domain, (**a**) the domain, (**b**) initial configuration and boundary condition, (**c**) the DONE, CLOSE, and FAR sets, and (**d**) partially computed distance field.

**Figure 3 materials-15-05376-f003:**
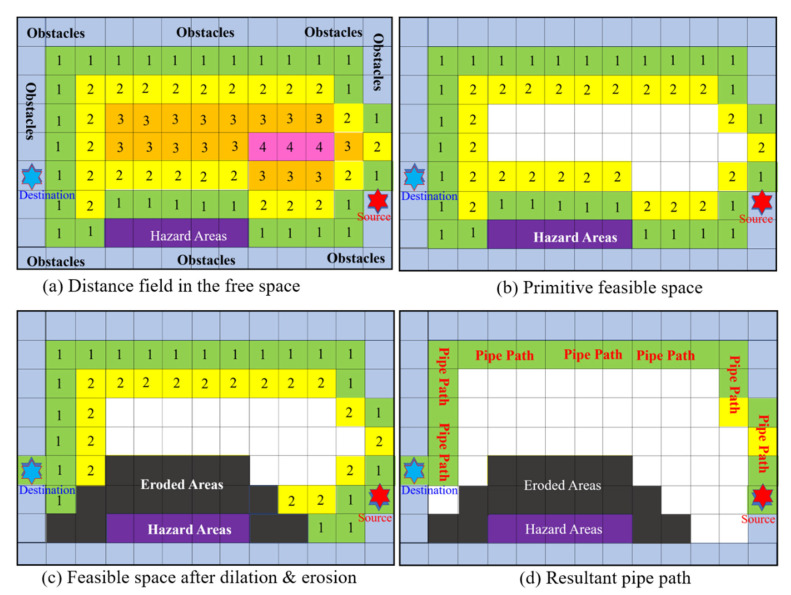
A 2D example illustrating the construction and variation of the feasible space, (**a**) the free space, (**b**) the primitive feasible space, (**c**) the final feasible space, (**d**) the pipe path composed of voxels.

**Figure 4 materials-15-05376-f004:**
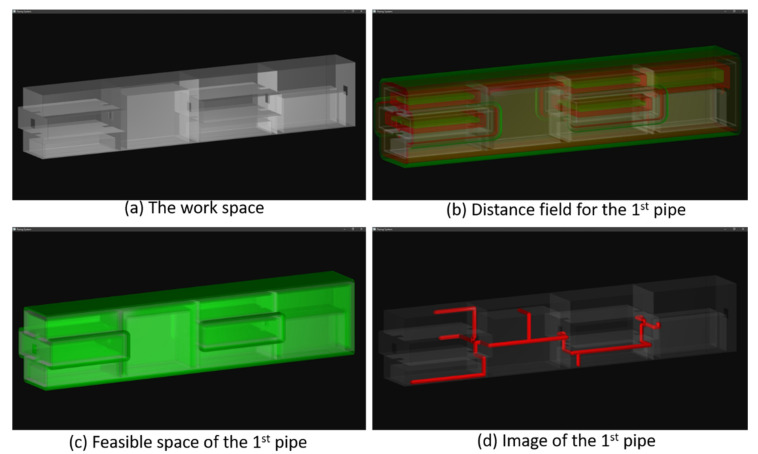
Routing the 1st pipe inside the underwater vehicle, (**a**) internal structure of the domain, (**b**) two levels in the distance field, shaded in green and red colors, (**c**) the feasible space, shaded in green color, and (**d**) the resultant pipe, shaded in red color.

**Figure 5 materials-15-05376-f005:**
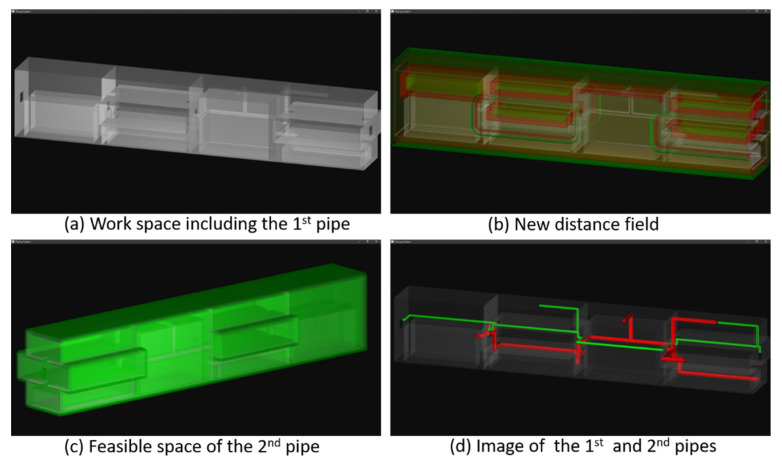
Routing the 2nd pipe, (**a**) the modified workspace, (**b**) two layers in the new distance field, shaded in green and red colors, (**c**) the feasible space of the 2nd pipe, and (**d**) the results, including the 1st and 2nd pipes (in red and green colors).

**Figure 6 materials-15-05376-f006:**
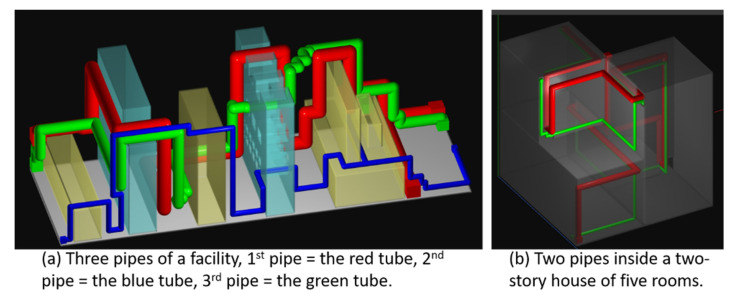
(**a**) Routing 3 pipes in a machinery room, (**b**) routing 2 pipes in a 2-story house.

**Table 1 materials-15-05376-t001:** Geometrical information of the workspace.

Size (Length × Width × Height)	8.02 × 2.00 × 2.00 (m)
Resolution of grid	802 × 200 × 200
Voxel size	0.01 × 0.01 × 0.01 (m)

**Table 2 materials-15-05376-t002:** Coordinates of the terminals of the 1st pipe.

Terminals	Coordinates
1	186, 102, 64
2	380, 32, 56
3	186, 102, 139
4	781, 32, 56
5	499, 155, 101
6	654, 96, 170
7	702, 170, 101

Pipe radius = 4 cm.

**Table 3 materials-15-05376-t003:** Coordinates of the terminals of the 2nd pipe.

Terminals	Coordinates
1	3, 108, 93
2	800, 108, 93
3	311, 172, 101
4	709, 170, 101

Pipe radius = 2 cm.

## Data Availability

Not applicable.

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
