# Peer review of "A Distance-Field-Based Pipe-Routing Method"

_materials, 2022, doi:10.3390/ma15155376_

Round 1

Reviewer 1 Report

The paper proposes a distance field-based piping algorithm that optimizes the pipe costs while meeting the spatial constraints and safety regulations. In general, the paper is well-written with clear originality, detailed methodology, and well-presented results. No wonder why it is selected among the excellent papers in iTIKI IEEE ICASI 2021. 

There are only Minor issues that should be addressed to further improve the paper particularly on the Discussion part.

1. The significance of the study should be identified by discussing the applications in the real world setting. This should support the statement in the Abstract: "Pipe routing is essential in many industrial applications, including ship-building, engine design, electric circuit layout, etc."

2. Also, discuss the novelty of the findings relative to existing studies. How the proposed method is better in terms of cost minimization, while  satisfying the spatial constraints and safety regulations? What new insights we can learn from the findings?

3. The references must be improved significantly by citing the most recent (preferable 5y or newer) and relevant studies. Focus on the Literature Review of most recent studies as well as in the Discussion section in connection to comment #2.

4. Figures (4, 5, 6b) should be improved by making the 3D pipes as distinguishable as Fig. 6a. Also, increase the font size to make it readable (e.g. Fig. 2, 3) 

Author Response

  1. The significance of the study should be identified by discussing the applications in the real world setting. This should support the statement in the Abstract: "Pipe routing is essential in many industrial applications, including ship-building, engine design, electric circuit layout, etc."

Reply: We added extra materials in the test results to explain why pipe routing is essential in the designs of underwater vehicles, machinery rooms, and buildings. These contexts are contained in Sections 4.1 and 4.2 and printed in red-color. The first paragraph of the Abstract is also slightly modified to clarify the importance of pipe routing.

  1. Also, discuss the novelty of the findings relative to existing studies. How the proposed method is better in terms of cost minimization, while satisfying the spatial constraints and safety regulations? What new insights we can learn from the findings?

Reply: We rewrite the first paragraph of the Discussion section and itemize the advantages of the proposed method. Please, see Section 5 for the contents.

  1. The references must be improved significantly by citing the most recent (preferable 5y or newer) and relevant studies. Focus on the Literature Review of most recent studies as well as in the Discussion section in connection to comment #2.

Reply: We has added the recommend references in the manuscript and describe their contributions and applications in the “Related Work” and “Discussion” sections (Sections 2 and 5). We also make brief comparison between their methods and ours.

  1. Figures (4, 5, 6b) should be improved by making the 3D pipes as distinguishable as Fig. 6a. Also, increase the font size to make it readable (e.g. Fig. 2, 3)

Reply: Figures 4, 5, and 6b are produced by using surface rendering and volume rendering. The workspaces are displayed by using volume rendering such that we can see through them and find the pipes. If we remove some of the walls, floors, and ceilings, the pipes will be better displayed. However, the cue of the relative positions of the pipes and the distances between the pipes and the obstacles will be lost. Thus, we keep the walls, floors, and ceilings when rendering the images. By combining surface rendering and volume rendering to illustrate the pipes and the workspace helps us to comprehend the internal structures of the workspaces as well as the topologies of the pipes. Furthermore, volume rendering improves the “depth cue” of the pipes and makes the images more three-dimensional. Extra contexts are added in Section 3.5 to explain our rendering methodology.

   The fonts of Figures 2 and 3 have been enlarged to improve the image qualities. Thanks for the suggestions.

Reviewer 2 Report

In this paper, the authors propose a distance-field-based piping algorithm, where the costs of the resultant pipes are optimized, the spatial constraints and safety regulations are strictly obeyed by the pipes, and the progression of the piping process is revealed in real-time. 

This paper presents an original contribution. The abstract is sufficient to give useful information about the paper’s topic. The proposed technique is somehow described and thoroughly illustrated. Results need more explanation and justification. Finally, this paper is somehow well structured. However, there are some comments we recommend the authors to do:

 In the introduction section or where appropriate, you may need to cite the following references, regarding the new pipe routing approach, 3-D space constraints, and A*: 

A New Pipe Routing Approach for Aero-Engines by Octree Modeling and Modified Max-Min Ant System Optimization Algorithm. Journal of Mechanics, 34(1), 11–19, 2018. 

Integral Layout Optimization of Subsea Production Control System Considering Three-Dimensional Space Constraint. Processes, 9(11), 1947, 2021. 

Performance evaluation of parallel multithreaded A* heuristic search algorithm. Journal of Information Science, 40(3), 363–375, 2014.

In Section 2, you need to present your proposed routing algorithm and other algorithms you did use as pseudocode and accordingly explain them in detail.

In Section 3 (Results), if possible, compare your approach with other existing approaches such as the Genetic Algorithm.

Also, in Section 3, you need to evaluate your algorithm (approach) in terms of accuracy and runtime to check the trade-off between runtime and accuracy of the proposed approach in comparison with the other approaches or algorithms.

In Section 5 (Conclusions), you may need to change the title of this section to “Conclusion and future work” to expand your future work as a separate paragraph. However, in the future paragraph, you may mention also, that the mentioned problem can be solved using IDA* for large network pipes, where you may need to cite the following reference regarding IDA*: Parallel multithreaded IDA* heuristic search: Algorithm design and performance evaluation. International Journal of Parallel, Emergent and Distributed Systems, 26(1), 61–82, 2011.

General comments:

·       You may need to delete the title “1.1 Related work” and keep this section as part of the introduction.

·  Need to write one small overview paragraph in Section 2 before Subsection 2.1 about its subsections.

·     Please check the whole paper for formatting issues. Follow the journal’s guidelines.

Author Response

Our responses towards the second reviewer’s comments are presented in this file. The revisions of the manuscript are listed item-by-item. The reviewer’s comments and our responses are typed in black and red fonts respectively. We have tried our best to revise the manuscript according to the reviewer’ comments. We would like to express our appreciation toward the editors, the reviewers and anyone who help to process this article.

   (The structure of the manuscript has been modified according to the suggestion of the 2nd reviewer. The “Related Work” subsection becomes an independent section, and the indices of the following sections are increased by 1.)

Reviewer 2:

This paper presents an original contribution. The abstract is sufficient to give useful information about the paper’s topic. The proposed technique is somehow described and thoroughly illustrated. Results need more explanation and justification. Finally, this paper is somehow well structured. However, there are some comments we recommend the authors to do:

  • In the introduction section or where appropriate, you may need to cite the following references, regarding the new pipe routing approach, 3-D space constraints, and A*:
    • A New Pipe Routing Approach for Aero-Engines by Octree Modeling and Modified Max-Min Ant System Optimization Algorithm. Journal of Mechanics, 34(1), 11–19, 2018.
    • Integral Layout Optimization of Subsea Production Control System Considering Three-Dimensional Space Constraint. Processes, 9(11), 1947, 2021.
    • Performance evaluation of parallel multithreaded A* heuristic search algorithm. Journal of Information Science, 40(3), 363–375, 2014.

Reply: These papers have been added into the reference list. Their contents are also reviewed or compared with our work in the “Related Work” and “Discussion” sections.

  • In Section 2, you need to present your proposed routing algorithm and other algorithms you did use as pseudocode and accordingly explain them in detail.

Reply: The pseudo-codes of the proposed piping algorithm are listed in Appendix A. The method for computing the distance field is presented in Appendix B.

  • In Section 3 (Results), if possible, compare your approach with other existing approaches such as the Genetic Algorithm.

Reply: We make comparisons between other researchers’ work and ours. The contents are in Section 2 (the related work, a new section), Section 5, the “Discussion” section, and Section 6 (the conclusion-and-future-work section.) We suggest to use these AI methods for schedule pipe-routing process involving multiple branches and multiple pipes and to use our method for computing the paths of individual pipes to satisfy the spatial constraints and safety regulations. The IDA* methods would be implemented if the problem domain is large, but the general methodology of the proposed piping method will be preserved.

  • Also, in Section 3, you need to evaluate your algorithm (approach) in terms of accuracy and runtime to check the trade-off between runtime and accuracy of the proposed approach in comparison with the other approaches or algorithms.

Reply: The worst-case time complexity of the proposed piping algorithm is presented in Section 5. The accuracy of the spatial gaps is linearly depending on the resolution of the underlying grid. This property is explained in the 2nd last paragraph of Section 5 (the Discussion section.)

  • In Section 5 (Conclusions), you may need to change the title of this section to “Conclusion and future work” to expand your future work as a separate paragraph.

Reply: This is a good idea. We change the title as suggested.

  • However, in the future paragraph, you may mention also, that the mentioned problem can be solved using IDA* for large network pipes, where you may need to cite the following reference regarding IDA*: Parallel multithreaded IDA* heuristic search: Algorithm design and performance evaluation. International Journal of Parallel, Emergent and Distributed Systems, 26(1), 61–82, 2011.

Reply: We study several IDA* algorithms and make comparisons between them and Dijkstra’s and A* methods. These IDA* algorithms use less memory and can help us to alleviate our memory-usage problems. The related materials of the comparisons and discussion are presented in Section 5 and 6 (the “Discussion” and “Conclusion and Future Work” sections.)

  • You may need to delete the title “1.1 Related work” and keep this section as part of the introduction.

Reply: We make it as an independent section such that the “Introduction” section is less lengthy and easier to read.

  • Need to write one small overview paragraph in Section 2 before Subsection 2.1 about its subsections.

Reply: A paragraph as been added to briefly describe the contents of this section (re-indexed as Section 3.)

  • Please check the whole paper for formatting issues. Follow the journal’s guidelines.

Reply: We has down-loaded the sample-manuscript from the web-page of the journal and follow the journal’s style to prepare the manuscript. Hope it will fit the required format.

Round 2

Reviewer 2 Report

The authors did most comments. So, no more comments and modifications are required from my side.

Author Response

Thank you for your efforts on reviewing our manuscript.